# Sweet Cherry Plants Prioritize Their Response to Cope with Summer Drought, Overshadowing the Defense Response to *Pseudomonas syringae* pv. *syringae*

**DOI:** 10.3390/plants13131737

**Published:** 2024-06-24

**Authors:** Luis Villalobos-González, Claudia Carreras, María Francisca Beltrán, Franco Figueroa, Carlos Rubilar-Hernández, Ismael Opazo, Guillermo Toro, Ariel Salvatierra, Boris Sagredo, Lorena Pizarro, Nicola Fiore, Manuel Pinto, Vicent Arbona, Aurelio Gómez-Cadenas, Paula Pimentel

**Affiliations:** 1Centro de Estudios Avanzados en Fruticultura (CEAF), Rengo 2940000, Chile; luisvillalobosg1@gmail.com (L.V.-G.); iopazo@ceaf.cl (I.O.); gtoro@ceaf.cl (G.T.); asalvatierra@ceaf.cl (A.S.); 2Facultad de Ciencias Agronómicas, Departamento de Sanidad Vegetal, Universidad de Chile, La Pintana 8820808, Chile; claudia.carreras@gmail.com (C.C.); fran.ibv24@gmail.com (M.F.B.); nfiore@uchile.cl (N.F.); 3Programa de Doctorado en Ciencias Silvoagropecuaria y Veterinarias, Campus Sur, Universidad de Chile, La Pintana 8820808, Chile; 4Instituto de Ciencias Agroalimentarias, Animales y Ambientales, Universidad de O’Higgins, San Fernando 3070000, Chile; franco.figueroa@uoh.cl (F.F.); carlos.rubilar@uoh.cl (C.R.-H.); lorena.pizarro@uoh.cl (L.P.); manuel.pinto@uoh.cl (M.P.); 5Instituto de Investigaciones Agropecuarias INIA Rayentué, Rengo 2940000, Chile; bsagredo@inia.cl; 6Centro de Biología de Sistemas para el Estudio de Comunidades Extremófilas de Relaves Mineros (SYSTEMIX), Universidad de O’Higgins, Rancagua 2820000, Chile; 7Department Ciències Agràries i del Medi Natural, Universitat de Jaume I, 12071 Castellon de la Plana, Spain; vicente.arbona@camn.uji.es (V.A.); aurelio.gomez@uji.es (A.G.-C.)

**Keywords:** gas exchange, water relations, ABA and SA, *Pss*, *Prunus avium* L.

## Abstract

Disease severity and drought due to climate change present significant challenges to orchard productivity. This study examines the effects of spring inoculation with *Pseudomonas syringae pv. syringae* (*Pss*) on sweet cherry plants, cvs. Bing and Santina with varying defense responses, assessing plant growth, physiological variables (water potential, gas exchange, and plant hydraulic conductance), and the levels of abscisic acid (ABA) and salicylic acid (SA) under two summer irrigation levels. *Pss* inoculation elicited a more pronounced response in ‘Santina’ compared to ‘Bing’ at 14 days post-inoculation (dpi), and those plants inoculated with *Pss* exhibited a slower leaf growth and reduced transpiration compared to control plants during 60 dpi. During differential irrigations, leaf area was reduced 14% and 44% in *Pss* inoculated plants of ‘Bing’ and ‘Santina’ respectively, under well-watered (WW) conditions, without changes in plant water status or gas exchange. Conversely, water-deficit (WD) conditions led to gas exchange limitations and a 43% decrease in plant biomass compared to that under WW conditions, with no differences between inoculation treatments. ABA levels were lower under WW than under WD at 90 dpi, while SA levels were significantly higher in *Pss*-inoculated plants under WW conditions. These findings underscore the influence on plant growth during summer in sweet cherry cultivars that showed a differential response to *Pss* inoculations and how the relationship between ABA and SA changes in plant drought level responses.

## 1. Introduction

Crops must be able to respond to different stimuli throughout the growth and development process; also, they experience increased pathogen pressure due to reduced biodiversity in agricultural systems compared to that in natural environments [1]. Additionally, they depend on fertilizers and freshwater availability [2,3]. Plant–pathogen interactions in adverse environmental conditions, such as drought, influenced by climate change, will eventually undermine the productivity and viability of crops such as wheat, rice, corn, and soybean [4] as well as fruit trees including as Prunus species [5].

Among deciduous fruit tree species, sweet cherry (*Prunus avium* L.) is cultivated worldwide in Mediterranean and temperate climates [6,7,8]. Two main factors have been identified as threats to cherry orchard production: (i) outbreaks of *Pseudomonas syringae* pv. *syringae* (*Pss*) during months of high humidity and low temperatures [9] and (ii) drought episodes that typically occur after harvest, during the summer season [10,11]. *Pss*, along with a group of *Pseudomonas* spp. and pathovars (*P*. spp. pvs.), is responsible for causing systemic infection leading to the death of young trees in nurseries and bacterial canker disease in older trees, which is one of the main constraints for *Prunus* spp. cultivation [9,12,13]. *Pss* exerts its greatest pathogen pressure in cold and wet environments as it facilitates the colonization of host tissues [14,15], whereas its prevalence declines to be negligible with rising temperatures, especially in dry environments [16]. 

Like other *P*. spp. pvs., *Pss* transfers phytotoxins to host cells during infection, which promotes tissue establishment, suppressing the immune response and evading plant defenses [17]. Phytotoxins dampen the pattern-triggered immunity (PTI) signaling pathway and reduce reactive oxygen species (ROS) production for plant defense [18]. To counteract the phytotoxins, effector-triggered immunity (ETI) aims to restore PTI and enhance the defense response of the plant [19]. Furthermore, when the defense signal triggered by PTI and ETI exceeds a threshold, it can elicit a programmed plant cell death known as a hypersensitive response (HR) [20]. 

In the short term, HR confines the pathogen to the site of infection and provides subsequent resistance to disease for the plant [21,22]. In the long-term, HR promotes a systemic defense response induced by SA-dependent signaling, spreading to distant tissues in a process called “systemic acquired resistance”, SAR [22,23]. SAR provides long-lasting resistance against biotrophic pathogens, preparing the plant to resist subsequent infections [23,24]. However, it remains unclear how long SAR can be sustained and to what extent the trade-off effect on plant growth is induced by the immune response, particularly in long-living trees [25,26,27,28,29].

The advent of summer leads to a decrease in the likelihood of *Pss* outbreaks, but drought becomes the main constraint for carbon acquisition in sweet cherry orchards [30,31]. Severe water stress has adverse effects on growth, bud development and differentiation, and the accumulation of reserves that are essential to support the onset of the following season [10,11,32,33]. In response to reduced water availability, plants close their stomata to prevent excessive water loss through the leaves. This action prolongs the time it takes for plant organs to reach a water potential threshold that can cause further damage. Consequently, this mechanism helps preserve the plant hydraulic functionality as water stress increases. However, the decrease in stomatal conductance intrinsically reduces CO_2_ uptake at carboxylation sites, leading to a subsequent decrease in photosynthesis [34,35]. 

Abscisic acid (ABA) plays a pivotal role in regulating the response of stomatal closure to water stress [36,37]. ABA and SA have exhibited antagonistic responses, potentially linked to the prioritization of differential biotic and abiotic stress responses [38]. For instance, the application of ABA has been shown to suppress basal immunity to biotrophic pathogens in rice [39], as well as to inhibit the SA-associated defense response in arabidopsis and tomato [40,41]. In contrast, the effect of SA on the response to abiotic stresses remains controversial. On the one hand, the application of SA analogs has been shown to reduce ABA-responsive and -biosynthesis genes after salt treatment [38]. Furthermore, citrus genotypes with higher levels of SA in their leaves were more susceptible to both drought and heat [42]. However, the application of SA analogs has been proposed to promote tolerance to water stress [43,44,45].

There is evidence of differential susceptibility to *Pss* between sweet cherry cultivars [13,46,47] dependent on the chemical response involved in plant immunity to *Ps* in plants based on both PTI and ETI [48], which ultimately generates SAR through the SA defense response [23]. The SA-induced defense response has deleterious effects on plant growth [28], but the extent of the defense response and its effect on growth remains elusive in fruit tree species and especially in sweet cherry varieties, especially on those with differential susceptibility to *Pss*. Additionally, the effects of the defense response on growth during the summer season in sweet cherry plants have not been thoroughly explored. This becomes more relevant given the decreasing pressure of *Pss* during the summer and the increased water requirements of the plants, and notably so in the scenario of water deficit conditions. Furthermore, the recently described antagonism between defense-related SA and the drought-response ABA might elucidate the prioritization of one response at the expense of the other [41,42,49]. 

This study explores the effects of spring inoculation with *Pseudomonas syringae pv. syringae* (*Pss*) using sweet cherry cultivars with a contrasting defense response to the pathogen on plant growth, water potential, stomatal conductance, photosynthesis, plant hydraulic conductance, abscisic acid (ABA), and salicylic acid (SA) when plants challenge two different levels of irrigation during the summer. For this purpose, we selected sweet cherry plants from the cvs. ‘Bing’, described as susceptible to *Pss* [46], and ‘Santina’, reported as tolerant [50]. More specifically, we analyzed how the effects of these stresses will lead to changes in plant growth, leaf water potential, gas exchange, plant hydraulic conductance, and final biomass. Also, we assessed the variations in ABA and SA levels in response to *Pss* spring inoculations in contrasting irrigation regimes during the summer. 

## 2. Results

### 2.1. Early Response to Inoculation with Pss in Sweet Cherry Plants

Necrosis, gum secretion, and apex death were observed in plants inoculated with *Pss* at 14 days post-inoculations (dpi), showing in a less-pronounced response in ‘Bing’ (Figure 1A, Appendix A) compared that in to ‘Santina’ (Figure 1B, Appendix A). The primary macroscopic responses to inoculation were classified into three levels: no response, gum secretion with necrotic wounds lower than 5 cm, and necrotic wounds higher than 5 cm, including apex death (Figure 1A,B, Appendix A). 

Plants of the ‘Santina’ variety showed a higher presence of macroscopic responses to *Pss* inoculation compared to those of ‘Bing’ (Table 1). Among the *Pss*-inoculated plants, 29% of the ‘Bing’ plants showed no response, whereas only 10% of the ‘Santina’ plants displayed a similar reaction (Table 1). Furthermore, within the same inoculation treatment, 62% of the ‘Bing’ plants exhibited local necrosis with gum secretion, while only 24% of the ‘Santina’ plants displayed these symptoms. In contrast, extended necrosis with apex death was observed in 10% of the ‘Bing’ plants, while 67% of the ‘Santina’ plants showcased this specific macroscopic response (Table 1). The control plants in both varieties showed healed wounds without visible gum secretion or necrosis (Table 1, Figure 1C, Appendix A).

*Pss* detection was confirmed by qPCR at 14 dpi analysis near the wound site in plants inoculated with the bacteria (Appendix A). On the same date, in stem samples beyond 5 cm from the inoculation site, *Pss* remained undetected in both varieties (Appendix A). In plants inoculated with the control, two of the six samples amplified bacterial DNA by qPCR, but at such a low level that its presence was negligible (Appendix A).

### 2.2. Effect of Inoculations Treatments on Leaf Growth Parameters

Between 50 and 65 dpi, plants inoculated with *Pss* needed more growing degree days (GDDs; °C day) for leaf emergence (phyllochron) and a longer duration of leaf expansion compared to that of the control plants. This led to a slower vegetative growth in plants inoculated with *Pss* compared to that in the control plants (Figure 2). The phyllochron was 22% higher in ‘Bing’ plants inoculated with *Pss* compared to those inoculated with the control, while in ‘Santina’, the difference between inoculation treatments was even greater, reaching 33% (Figure 2A). 

In general, ‘Bing’ showed a shorter thermal time for leaf expansion duration compared to ‘Santina’ (Figure 2B). Furthermore, in both varieties, plants inoculated with *Pss* required longer GDDs for leaf expansion duration compared to those inoculated with the control (Figure 2B). For ‘Bing’, the leaf expansion duration was 144 GDDs (CI 95%: 138–149) for plants inoculated with the control, while for those *Pss*-inoculated, it was 182 GDDs (CI 95%: 175–190) (Figure 3B). For ‘Santina’, the leaf expansion duration was 199 GDDs (CI 95%: 193–204) and 267 GDDs (CI 95%: 257–278) for plants in the control and Pss-inoculated, respectively (Figure 2B).

### 2.3. Effect of Spring Inoculations with Pss on the Plant Transpiration under Two Irrigation Regimes during Summer

Plant transpiration (Tr) increased in well-watered (WW) plants by between 70 and 93 dpi (Figure 3), but to a lesser extent in the case of ‘Santina’ plants inoculated with *Pss* during spring (Figure 3B). Also, Tr decreased in all plants subjected to water deficit (WD), regardless of the inoculation treatment and variety (Figure 3). 

Initially, Tr was 22% lower in *Pss*-inoculated plants compared to those in the control for both ‘Bing’ and ‘Santina’ until 67 dpi. From 70 dpi onward, ‘Bing’ plants under the WW irrigation regime showed a similar trend in Tr with a daily increase of 8.6 ± 1.1 mmol s^−1^ per day in the control plants and 9.7 ± 1.0 mmol s^−1^ per day in *Pss*-inoculated plants (Figure 3A, Appendix A). In contrast, in ‘Santina’ under the WW irrigation regime, Tr increased by 11.1 ± 1.1 mmol s^−1^ per day in the control plants, which was nearly twice than that observed in the *Pss*-inoculated plants, which had a rate of increase of 5.6 ± 1.0 mmol s^−1^ per day (Figure 3B, Appendix A). Also, with the WD irrigation regime, Tr decreased by an average of 4.3 mmol s^−1^ per day in both inoculation treatments in each variety (Figure 3, Appendix A).

### 2.4. Physiological Responses to Inoculation Treatments and Water Regimes of Plants during Summer

For both ‘Bing’ and ‘Santina’, the plant water status (Ψ_md_), stomatal conductance (g_sw_), and net assimilation (A_n_) responded to the irrigation regimes independently of the spring inoculation treatments (Figure 4). At the WW irrigation level, the midday water potential (Ψ_md_) of the ‘Bing’ (Figure 4A) and ‘Santina’ plants (Figure 4B) showed mean values between −0.72 and −0.84 MPa, while the plants at the WD irrigation level showed a decrease from −0.94 at 72 dpi to −1.77 MPa at 93 dpi.

The g_sw_ was higher than 0.15 mol H_2_O m^−2^ s^−1^ at the WW irrigation level for both ‘Bing’ (Figure 4C) and ‘Santina’ (Figure 4D). At the WD irrigation level at 72 dpi, the mean values of g_sw_ were 0.155 mol H_2_O m^−2^ s^−1^ and 0.093 mol H_2_O m^−2^ s^−1^ for ‘Bing’ (Figure 4C) and ‘Santina’ (Figure 4D), respectively. Later, from 78 to 93 dpi, the g_sw_ values dropped by 0.06 mol H_2_O m^−2^ s^−1^ on average for both varieties. Again, there were no differences between the plants with the spring inoculation treatments. 

The net assimilation rate (A_n_) showed similar values between inoculation treatments in both ‘Bing’ (Figure 4E) and ‘Santina’ (Figure 4F). Also, photosynthesis was 1.5 times higher under WW conditions compared to that in WD conditions at 72 dpi. From then on, this difference between WW and WD irrigation regimes increased five-fold. 

As shown in Figure 5, K_plant_ at 93 dpi showed higher values in WW plants compared to those in WD plants (F_1,39_ = 113, *p* = <0.001) without differences between varieties (F_1,39_ = 0.60, *p* = 0.441) and spring inoculation treatments (F_1,39_ = 0.21, *p* = 0.645). At the WW irrigation level, ‘Bing’ showed on average 2.7 ± 0.3 mmol H_2_O m^−2^ s^−1^ MPa^−1^, while in ‘Santina’, it was 2.3 ± 0.2 mmol H_2_O m^−2^ s^−1^ MPa^−1^. For the WD irrigation level, the mean values of K_plant_ were 0.40 ± 0.05 mmol H_2_O m^−2^ s^−1^ MPa^−1^ and 0.47 ± 0.07 mmol H_2_O m^−2^ s^−1^ MPa^−1^ for ‘Bing’ and ‘Santina’, respectively (Figure 4E). 

### 2.5. Salicylic and Abscisic Acids Content in Leaves

Abscisic acid (ABA) increased (Figure 6A), and salicylic acid (SA) decreased as the irrigation level was reduced (Figure 6B). Moreover, *Pss*-inoculated plants showed higher levels of SA at the WW irrigation level, particularly in ‘Santina’ compared to ‘Bing’ (Figure 6B). 

Regardless of the varieties, the levels of ABA were the lowest in control plants at the WW irrigation level, with mean values of 2.2 ± 0.2 µg g^−1^ in ‘Bing’ and 2.8 ± 0.2 µg g^−1^ in ‘Santina’, compared to *Pss*-inoculated plants showing slightly higher levels of ABA, with 3.2 ± 0.2 µg g^−1^ and 3.5 ± 0.2 µg g^−1^ for ‘Bing’ and ‘Santina’, respectively. On the contrary, the highest levels of ABA were observed in the control plants at the WD irrigation condition with values of 6.0 ± 0.9 µg g^−1^ for ‘Bing’ and 6.4 ± 0.5 µg g^−1^ for ‘Santina’, while those in *Pss*-inoculated plants were 5.0 ± 0.5 µg g^−1^ and 4.8 ± 0.5 µg g^−1^ for ‘Bing’ and ‘Santina’, respectively.

SA levels were higher in plants under WW irrigation conditions than in those under WD conditions. Additionally, *Pss*-inoculated plants showed increased SA levels compared to the control plants. In descending order, the mean SA values were 11.3 ± 1.0 µg kg^−1^ and 8.7 ± 1.0 µg kg^−1^ for ‘Santina’ and ‘Bing’, respectively, for *Pss*-inoculated plants under the WW irrigation condition. This was followed by 7.8 ± 0.5 µg kg^−1^ and 5.2 ± 0.9 µg g^−1^ for ‘Santina’ and ‘Bing’, respectively, in control plants at the WW irrigation level. For the WD irrigation condition, SA averaged 3.7 µg kg^−1^ in ‘Santina’ and 2.2 µg kg^−1^ in ‘Bing’ for both inoculation treatments.

### 2.6. Leaf Area and Plant Biomass at Harvest

The evaluation of the whole leaf area, the number of leaves, and the leaf abscission was conducted at the end of the experiment (Table 2). Leaf area showed different responses depending on the inoculation treatment at the respective irrigation regime (*Irri × Ino* = 0.0013). Regarding the WW irrigation, *Pss*-inoculated plants showed a decreased leaf area compared to that of the control plants. This reduction was more pronounced in ‘Santina’ (44% lower) than in ‘Bing’ (14% lower). Conversely, for both inoculation treatments subjected to the WD irrigation, the leaf area decreased by up to 40% when compared to the plants from the control under the WW irrigation regime (Table 2). 

In ‘Bing’ under the WW irrigation, the number of leaves was reduced by 25% in *Pss*-inoculated plants compared to that on the control plants. Furthermore, plants under WD conditions showed 25% fewer leaves in both inoculation treatments compared to control plants under WW irrigation. Additionally, the percentages of fallen leaves among the ‘Bing’ plants were similar (Table 2). 

In ‘Santina’ under the WW irrigation, the *Pss*-inoculated plants’ number of leaves decreased by 52% compared to the control but maintained a similar percentage of fallen leaves. Interestingly, both inoculation treatments at the WD irrigation level showed 45% fewer leaves. Furthermore, ‘Santina’ exhibited the highest percentage of leaf abscission, reaching 34% in both treatments at the WD irrigation level. This percentage is nearly twice as high as that observed in all other combinations of irrigation and inoculation between the varieties (Table 2).

Root and plant dry weight were reduced in plants under WD irrigation, as well as in *Pss*-inoculated plants under WW irrigation compared to their control plants at the WW irrigation level. Additionally, the final biomass of the root and plant in the ‘Santina’ variety showed higher responses than ‘Bing’ between inoculation treatments at the WW irrigation level (Table 2). Root and plant dry weight were 23% and 17% lower in *Pss* plants compared to the control plants in the WW irrigation regime in ‘Bing’, whereas ‘Santina’ showed higher reductions of 46% and 44% of the root and plant dry weight under the same irrigation conditions. Once again, in plants subjected to the WD irrigation regime, no differences were observed between the varieties and inoculation treatments, with mean values of 75 g for the root dry mass and 255 g for the plant biomass (Table 2).

## 3. Discussion

Sweet cherry trees have shown substantial variability in their sensitivity to *Pss* across different genotypes [13,46,51], as well as within a single genotype to different strains of the bacterium [52,53]. Visual manifestations of macroscopic responses, such as gum secretion and necrosis, have been reported after *Pss* inoculation through shoot wounding [47,54]. In this study, gum secretion and necrosis were evident at 14 dpi (Figure 1, Table 1), accompanied by the presence of bacteria near the inoculation site (Appendix A). However, the bacteria were not detected in healthy tissue beyond necrotic wounds at 14 dpi (Appendix A) or in leaves at 81 dpi (Appendix A). 

The extended necrosis, especially in ‘Santina’, and the absence of the mild pathogenic strain of *Pss* beyond it are related to a hypersensitivity response triggered by the plant. This defense mechanism is activated to hinder the development and the spread of the pathogen during the colonization of plant tissues by biotrophic pathogens [21,22,55]. Also, it involves a sustained heightened state of defense in the longer term, leading to a faster, stronger, and/or more sustained mobilization of cellular defenses than plants facing pathogens for the first time [25]. In this research, the removal of tissue near the inoculation site (that also was necrotic) introduces uncertainty regarding the extent of damage caused by longer exposure to *Pss*, particularly in ‘Bing’ plants; however, it is necessary to consider that the pressure of this pathogen decreases during the summer season until it becomes negligible [9,16]. 

We observed variations in plant growth between inoculation treatments prior to the implementation of irrigation levels between 50 and 65 dpi, as indicated by the phyllochron (Figure 2A) and the duration of leaf expansion (Figure 2B). Both parameters imply an initially slower canopy growth rate in *Pss*-inoculated plants compared to those in the control, with the differences being more pronounced in ‘Santina’ plants than in ‘Bing’ plants (Figure 2). Therefore, the observed increase in Tr throughout the experiment (Figure 3) is primarily attributed to the expanding leaf area of the plants under the WW irrigation regime, since reference evapotranspiration (ET_0_) ranged from 5 mm to 8 mm without a clear changing trend (Appendix A), showing a stable water demand throughout the experiment. Interestingly, the ‘Santina’ plants inoculated with *Pss* during spring exhibited the lowest daily increment in Tr during the season among all plants under the WW irrigation regime (Figure 3B, Appendix A). Remarkably, this lower trend in Tr was without any apparent limitations in plant water status (Figure 4B), stomatal function (Figure 4D), CO_2_ assimilation (Figure 4F), and plant hydraulic conductance (Figure 5). These findings confirm that the initial state of slower leaf growth (Figure 2) in this group of plants persisted until 90 dpi, resulting in a lower leaf growth rate. Such phenomena have been previously documented in plants that have a sustained heightened defense state, resulting in growth trade-offs in the absence of the pathogen [28,56,57,58].

Plant growth is tied to transpiration through its leaves [59,60,61]. However, no evidence was found that indicated any detrimental effect on water relations or photosynthesis (Figure 4 and Figure 5) that could explain the observed slower growth between plants inoculated with *Pss* and those in the control group under the WW irrigation regime, particularly in the case of ‘Santina’. Additionally, *Pss*-inoculated plants in the WW group resulted in lower leaf aera, without differences observed in the proportion of abscised leaves and lower final root and plant biomass (Table 2). Clearly, ‘Santina’ experienced a more pronounced reduction in plant growth compared to ‘Bing’. In addition, they had higher SA levels in their leaves at 90 dpi (Figure 6B). The slower plant growth during summer seen in ‘Santina’ inoculated with *Pss* and the higher SA levels in leaves could be evidence of the role of this SAR response, as previously observed in different plants challenged to infections with biotrophic pathogens [58,62]. Also, the defense responses that involve salicylic acid among others promote the metabolism of sulfur [63], nitrogen, and phosphorus [64,65]. This, in turn, affects the availability of these nutrients for other physiological processes such as growth. Additionally, the reduction in sulfur in necrotic zones triggered by hypersensitive response (HR) has been observed to influence this dynamic [66]. However, further studies are needed to explore nutrient metabolism and its effects on the growth/defense trade-off. Moreover, disease-resistant plants may incur yield penalties in the absence of the pathogen when compared to susceptible varieties [56,57]. Currently, there are ongoing efforts to mitigate these penalties through the application of new gene editing techniques [67,68,69]. Nonetheless, it has been anticipated that when disease outbreaks are uncertain, cultivating disease-resistant varieties may be more optimal because it reduces the likelihood of a disease outbreak occurring in the orchard [70].

Certainly, plants under the WD irrigation regime showed a decreasing trend in Tr during the summer (Figure 3), but they also exhibited lower Ψ_md_ (Figure 4A,B), g_sw_ (Figure 4C,D), A_n_ (Figure 4E,F), and K_plant_ (Figure 5) compared to those under the WW irrigation regime, irrespective of the inoculation treatments. Plants regulate transpiration by closing their stomata, preventing a substantial drop in water potential and reducing the conductance of CO_2_ to the carboxylation sites [71]. This avoids that tension in the water column exceeding a critical threshold which leads to a hydraulic failure and results in a decline in the canopy or the entire plant [72]. It is described that soil drought reduces g_sw_ through changes in leaf turgor and ABA content [37,73]. As expected, plants under the WD irrigation regime with a reduced gs were accompanied by an increase in the ABA content at 90 dpi in leaves compared to those under the WW irrigation condition (Figure 6A).

In general, plant growth is significantly reliant on photosynthesis, which efficiently converts light energy into CO_2_ fixation, ultimately leading to biomass accumulation throughout the growing seasons [74]. We observed that plants under the WD irrigation regime declined in their overall growth, showing a reduced leaf area and lower root and total biomass compared to control plants under the WW irrigation regime (Table 2) but irrespective of the inoculation treatment in plants under drought. The lack of growth differences between inoculation treatments could be related to the antagonistic relationship between abscisic acid (ABA) and salicylic acid (SA). This antagonism may result in prioritizing a response to drought over maintaining defense mechanisms against new pathogen attacks [26]. Further experiments including inoculation trials in the post-drought period are required to deepen this hypothesis and re-evaluate the plant defense response in a longer term.

### Limitations of the Study and Future Work

In this study, one-year-old plants subjected to inoculations with a medium pathogenicity strain of *Pss* were used. Furthermore, the removal of tissue near the inoculation site (also that which was necrotic) introduces uncertainty as to the extent of damage caused by longer exposure to *Pss* with a strain of higher pathogenicity and/or the use of older plants. In future experiments, it would be useful to analyze whether sweet cherry cultivars exhibit altered responses to *Pss* when previously exposed to drought conditions.

## 4. Materials and Methods

### 4.1. Plant Material

The experiment was carried out at the Centro de Estudios Avanzados en Fruticultura (CEAF, 34°19′21′ S; 70°50′02′ W) using one-year-old plants of the sweet cherry cvs. ‘Bing’ and ‘Santina’ grafted on ‘Gisela 12’ rootstocks (*P*. *cerasus* × *P*. *canescens).* Rootstocks were obtained from Agromillora in April 2020, and were grown in 10 L transplant bags in a shade cloth nursery until March 2021. The buds of ‘Bing’ and ‘Santina’ were collected from two different locations: the commercial orchard Agrícola Amalfi (Codegua, O’Higgins region, Chile) and the Centro de Evaluación Rosario (Rengo, O’Higgins region, Chile), respectively. Sweet cherry varieties were confirmed by microsatellite marker analysis [75]. 

Two dormant buds of each variety were grafted 20 cm from the substrate. Thirty plants for each variety ‘Bing’ and ‘Santina’ were used. At the beginning of the dormancy of the plant, irrigation was stopped. After the sprout, only one grafted shoot per plant was maintained. Then, plants were inoculated with a strain of *Pss* on 16 November 2021, as described below. 

### 4.2. Bacterial Strain and Plant Inoculation

The bacterial strain used was PssA1M3, which has been described as mildly pathogenic, and it was isolated from a commercial sweet cherry orchard in the Ñuble region. The bacterial inoculum was prepared from a pure stock of the isolated strain and stored at −80 °C in Lysogeny broth (LB) medium. A portion of the culture was taken and grown in Petri dishes containing Pseudomonas agar F (PAF) medium supplemented with cycloheximide at 100 µg mL^−1^. The Petri dishes were incubated at 26 °C for 16 h. After incubation, a single colony was transferred to a tube containing liquid LB medium. The tube was incubated at 26 °C with shaking at 100 rpm for 12 h, promoting an exponential phase of the bacterial culture. 

On the day the plants were inoculated, an aliquot of the bacterial suspension was transferred to a bottle containing fresh LB medium. The bottle was incubated at 26 °C and shaken at 100 rpm until the suspension reached 0.1 O.D. at 600 nm. The bacterial suspension was precipitated by centrifugation at 3500× *g* and resuspended in sterile distilled water. The bacterial inoculum was, on average, 10^8^ CFU mL^−1^. 

For each variety, the plants were divided into two subgroups of 15 individuals. One subgroup was inoculated with *Pss*, while the other with sterile water as a control. The inoculation was carried out in a greenhouse on 16 November 2021. For each plant, the shoot was cut between the third and sixth internodes from the apex to the base. Sterile razor blades were used to wound at a 45° angle, deepening them to half the diameter of the shoot. Immediately, 50 µL of inoculum was added to the wound, followed by 2 drops of glycerol in the surrounding area. The wound was then covered with parafilm to create a local moist chamber, provide stem support, and bind the tissue together for proper wound healing. 

### 4.3. Evaluation of Plant Response to Pss and Experimental Design in the Field

Then, plant responses to inoculation treatment were evaluated at 14 days after inoculation (dpi). Plant responses included observation of visible gum secretion, necrotic tissue, and apex death. Also, the plants were analyzed by qPCR to detect the presence of *Pss* (Appendix A). For this purpose, three random plants were sampled from each inoculation treatment within each variety, collecting the first 2 cm and 5 to 10 cm below the inoculation site of the stem. The remaining 12 plants of each inoculation treatment within each variety were moved from the greenhouse to the shade cloth nursery, keeping them separated between each inoculation treatment. At 15 dpi, all plants were decapitated below the inoculation site, taking care to remove all necrotic tissue to prevent necrotrophic pathogens from thriving [76] and ensure plant recovery.

At 25 dpi, the plants were transplanted into 20 L pots, each filled with approximately 15 kg of substrate at 25 dpi. The substrate was a mixture of 1:1 peat and perlite supplemented with a 6 g L^−1^ Basacote Plus 9M (COMPO EXPERT GmbH, Münster, Germany) as a controlled-release fertilizer. In addition, the pots were covered with plastic bags to prevent soil evaporation. One to three apical buds were allowed to sprout and left to grow for 30 days before the plants were transported to field conditions (full sunlight). Finally, 48 plants from each inoculation treatment, 12 control and 12 *Pss* within each variety (‘Bing’ and ‘Santina’), were transported to the field on December 28. The plants were distributed in 6 blocks, and each block contained 8 plants corresponding to the 2 irrigation regimes (well-watered, WW and water deficit, WD) for each inoculation treatment (control and *Pss*) within each variety (‘Bing’ and ‘Santina’). Each block represented a replicate, each plant was defined as an experimental unit, and pots were randomly distributed within each block. All plants were irrigated three times a week for about 45 min using two drippers per plant with a flow rate of 2 L h^−1^ until differential irrigation regimes were implemented.

### 4.4. Irrigation Regimes

Irrigation regimes were started on January 19th (64 dpi). Fifteen days before the establishment of irrigation regimes, each pot was watered until saturation, and its weight was recorded when water drainage stopped. The recorded weights for each pot were averaged, and the value is referred to as the weight at field capacity (W_fc_). From 64 dpi, the plants were watered three times a week. Before each watering, the weight of each pot was measured using a 30 kg digital weighing scale with an accuracy of 50 g. Then, for each pot, the water transpired (Ti) was determined as the difference between the weight after irrigation and its weight in the morning before the next irrigation.

In the WW irrigation regime, the amount of water for each plant was completely replenished until soil saturation. Meanwhile, in the WD irrigation regime, water was manually replenished to reach the fraction of transpirable soil water (FTSW) threshold of the plant with the fourth-lowest Ti. This continued until all plants in the WD treatment reached a FTSW value of 0.4. From then on, the water was replenished to maintain the FTSW values between 0.4 and 0.2 until the plants were harvested.

The fraction of transpirable soil water (FTSW) was calculated according to Sinclair and Ludlow [77] as follows: $$FTSW=\frac{\left(T_{i}-W_{wp}\right)}{\left(W_{fc}-W_{wp}\right)}$$
where T*_i_* and W*_fc_* denote the amount of water transpired for the plant and the weight of each pot at field capacity, respectively. W*_wp_* refers to the weight of each pot at the wilting point. The W*_wp_* was obtained as 0.47 W*_fc_*. This ratio was determined based on the weight at which the leaves showed visible signs of wilting on a separate batch of 12 plants. 

### 4.5. Leaf Growth Parameters

The phyllochron (degree days required for leaf emergence) and the leaf expansion duration were evaluated from 50 dpi to 65 dpi, before the onset of irrigation treatments. They were expressed as the sum of the growing degree days (GDD) using 10 °C as the base temperature from INIA Rayentué Agrometeorological station (Appendix A). The growth dynamics of the leaves were analyzed in one shoot per plant from 50 to 65 dpi. For this purpose, the shoot near the apex, which was completely exposed to sunlight, was selected. The phyllochron was defined as the degree days required for leaf emergence, and it was calculated as the inverse of the slope from the linear regression between the number of new metamers and the accumulated thermal time [78]. Then, a regular phyllochron was assumed for all leaves on the same shoot. The leaf expansion duration was calculated as the sum of GDDs from the observed date of metamer appearance to the date of final size acquisition of each leaf in a pair. 

### 4.6. Physiological Variables

#### Plant Water Status and Whole Plant Hydraulic Conductance

The leaf water potential (Ψ) was assessed weekly from 69 to 90 dpi. The Ψ was measured using a pressure chamber (Model 1505D, PMS Instrument Company, Albany, OR, USA) on fully expanded, undamaged leaves located between nodes 7 and 15 from apex to base and fully exposed to the sun. Also, measurement was aided with a stereo microscope (Model: XTL-101Bled, L&T Optics, Fuqing, China) to visualize the sap emergence at the cut end of the petiole. Before measurement, leaves were wrapped in damp paper towels and enclosed in aluminized plastic bags for at least 1 h in the plant. Then, leaves were removed using razor blades, transported in a cooler box, and pressurized within the first five minutes after detachment. The leaves were placed into a chamber with the petiole protruding from the chamber lid. Nitrogen gas was used to pressurize the chamber, and the pressure required for the emergence of xylem sap was recorded. The midday Ψ_md_ was evaluated between 11:15 h and 13:45 h. Additionally, the pre-dawn water potential (Ψ_pd_) was measured between 5:00 h and 7:00 h only at 90 dpi. 

Hydraulic conductance of the whole (K_plant_) plant was determined using the following formula from Tyree and Zimmermann [79]:$$K_{plant}=\frac{∆E}{\left(\Psi_{pd}-\Psi_{md}\right)}$$
where ΔE is the change in the weight of the pot between predawn and midday.

### 4.7. Leaf Gas Exchange

Leaf gas exchange and chlorophyll content were evaluated weekly from 69 to 90 dpi. Leaf gas exchange was conducted 2–4 min before sampling the Ψ_mds_ (11:15 h and 13:45 h) using a portable gas analyzer (CIRAS-3, PP Systems Co., Ltd., Amesbury, MA, USA). Fully sun-exposed mature leaves with an incident PAR over 1400 µmol photons m^−2^ s^−1^, close to those selected for Ψ_mds_, were chosen for gas exchange measurement. To determine photosynthesis (A_n_) and stomatal conductance (g_sw_), the leaves were acclimated in the cuvette for 60 to 90 s until steady state was achieved. CO_2_ was adjusted to 400 µmol mol^−1^. The light, humidity, and temperature were set to ambient. 

### 4.8. Morphoanatomical Measurements

The leaves, internodes, and number of branches on each shoot were counted at the end of the experiment. Ten leaves per plant were scanned and dried in an oven for 2 days at 70 °C to determine their specific leaf area (sla). Also, the remaining leaves of each plant were dried as explained above, and their weight was used to calculate the final leaf area using the sla obtained earlier. 

### 4.9. Determination of ABA and SA

Phytohormones were evaluated in mature leaves collected 90 dpi, as previously described in both Durgbanshi et al. and Zandalinas et al. [42,80] with modifications. A total of 15 mg of ground and lyophilized tissue was spiked with 25 µL of [^2^H_6_]-ABA and [^13^C_6_]-SA at 1 mg L^−1^ as internal standards. Then, two glass beads were added to the sample. Hormones were extracted in 1 mL of mili-Q grade and homogenized in a bead beater for 10 min at a frequency of 15 Hz. The samples were centrifuged at 10,000× *g* for 10 min at 4 °C, and the precipitate was discarded. The resulting supernatants were adjusted to pH 3 with a 50 µL acetic acid 30%. Liquid:liquid extraction was performed twice using 1 mL of diethyl ether solution, and organic phase was collected. The organic solvent was evaporated to dryness in a speed-vac at room temperature. The dried residues were reconstituted in 0.5 mL of methanol 10%, and then, the extracts were 0.25× diluted with mili-Q water, filtered through 0.2 µm PTFE syringe filter membranes, and finally injected into the UPLC system (Acquity, Waters, USA) coupled to ESI-MS/MS (Xevo TQS, USA). The separation method [80] was carried out on a reverse-phase C18 column (Luna Omega C18 Polar, 5 cm × 0.2 cm i.d., 1.8 µm). The elution gradient was as follows: solvent A (Acetonitrile plus acetic acid 0.1%), solvent B (H_2_O HPLC grade with acetic acid 0.1%) with the following proportions: 0 min, 10% A and 90% B; 0–5 min, 90% A and 10% B; and 5–7 min, 10% A and 90% B with a flow of 0.3 mL min^−1^. ABA and SA were quantified using a triple quadrupole mass spectrometer (Xevo TQS, Waters, USA) operating in negative mode. Nitrogen was used for nebulization and drying gas at 350 °C and at flow of 60 L h^−1^ and 700 L h^−1^, respectively. Source temperature was set at 120 °C. Chromatogram data were processed with MassLinx V4.2 software (Waters, USA).

### 4.10. Statistical Analysis

All graph and statistical analyses were performed with the R software (version 4.2.2, R Core Team 2022). A linear mixed model [81] was used to test the effects of inoculation, irrigation, and interactions between them to evaluate the response of Tr, Ψ_md_, g_sw_, A_n_, and chlorophyll content in each variety on each day. The model was as follows:Y = µ + R_i_ +I_j_ + RI_ij_ + b_k_ + ε_ijkl_
where Y refers to the response variable for a plant, the µ is the general mean, R_i_ is the effect of irrigation (WW, WD), I_j_ is the effect of inoculation (control, *Pss*), RI_ij_ is the interaction between irrigation and inoculation, b_k_ is the random effect of the block, and ε_ijkl_ is the residual errors. 

A second linear mixed model was used to evaluate the response of K_plant_, leaves, root, shoot, plant characteristics, and phytohormones. The model test for effects of cultivars, inoculation, and irrigation and the interaction between them is as follows:Y = µ + V_i_ + R_j_ + I_k_ + VR_ij_ + VI_ik_ + RI_jk_ + VRI_ijk_ + b_l_ + ε_ijklm_
where Y refers to the response variable for a plant, the µ is the general mean, V_i_ is the effect of the sweet cherry cultivar (‘Bing’, ‘Santina’), R_j_ is the effect of irrigation (WW, WD), I_k_ is the effect of inoculation (control, *Pss*), VR_ij_ is the effect of interaction between cultivar and irrigation, VI_ik_ is the effect of interaction between cultivar and inoculation, RI_jk_ is the effect of interaction between irrigation and inoculation, VRI_ijk_ is the effect of interactions between cultivar, irrigation, and inoculation, b_l_ is the random effect of the block, and ε_ijkl_ is the residual errors. 

Linear mixed models were performed using the *lme* function [81]. Models were tested to fulfill the assumptions of normality and variance homogeneity. Also, models were tested using the Akaike information criterion (AIC) and Bayesian information criterion (BIC) for non-significant fixed effects (*p*-value > 0.05) and were removed to achieve the most parsimonious model.

A general linear model was used to evaluate the response of the number of leaves and the fallen leaves (%) to variety, inoculation treatments, and irrigation regimes. The models were adjusted to quasi-poisson and quasi-binomial errors for the leaf number and abscission leaf responses, respectively, using the *glm* function.

The paired comparisons and confidence intervals between parameters of linear and non-linear regressions were obtained using the package ‘emmeans’ by performing a post hoc Tukey test of marginal means [82].

## 5. Conclusions

We conclude that the amplitude of the early defensive response to *Pss* inoculation in ‘Santina’ results in more substantial growth detriment during summer (periods when the pressure of the pathogen is reduced and/or absent) compared to that in ‘Bing’ under a well-watered irrigation regime. 

The variations in growth under non-limiting water conditions were not attributed to constraints in plant hydraulics or CO_2_ assimilation. Instead, they were a consequence of a plant defense response that persisted under well-watered (WW) conditions, likely linked to elevated SA levels at 90 dpi in *Pss*-inoculated plants. This extended defense response may be related to SAR, as previously observed in different plants challenged to infections with biotrophic pathogens.

Conversely, under water-deficit irrigation regimes, plants’ ABA levels increase while SA levels decrease. This antagonistic interaction between these two hormones enables plants to prioritize their response to cope with drought, potentially overshadowing the extended defense state. Interestingly, the growth penalty observed in *Pss*-inoculated plants compared to those treated with the control under the WW irrigation regime, especially in ‘Santina’, was no longer evident under the water-deficit irrigation condition.

The research carried out during this study will offer critical information for both growers and nurseries on the selection of sweet cherry cultivars, the suitability of disease-free material, and the adverse effects on plants even when transiently affected by the Pss. This is particularly relevant during the growth phase in young sweet cherry plants.

## Figures and Tables

**Figure 1 plants-13-01737-f001:**
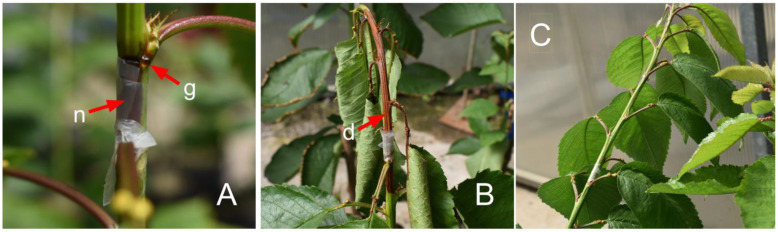
Macroscopic responses of sweet cherry plants to *Pss* inoculation. Photographs of lesions of shoots inoculated with *Pss* at 14 dpi on ‘Bing’ (**A**), ‘Santina’ (**B**), and control of Santina (**C**). The red arrows denote the macroscopic responses observed, including (**g**) gum secretion, (**n**) local necrosis (<5 cm), and (**d**) extended necrosis (>5 cm) with apex death.

**Figure 2 plants-13-01737-f002:**
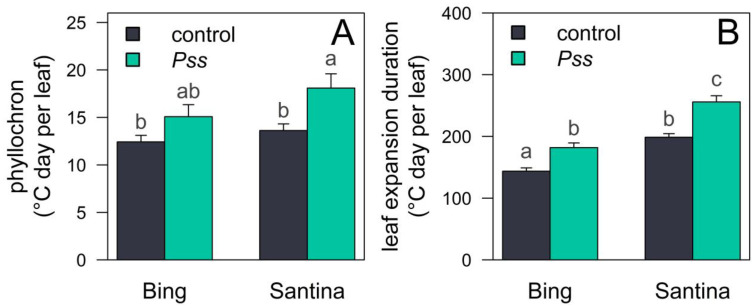
Response of leaf growth parameters in sweet cherry plants under inoculation treatments. (**A**) Phyllochron and (**B**) leaf expansion duration observed in ‘Bing’ and ‘Santina’ plants inoculated with the control (black) and *Pss* (green) during spring. The bars indicate means and SE (n = 6). Different letters denote significant differences between varieties and inoculation treatments.

**Figure 3 plants-13-01737-f003:**
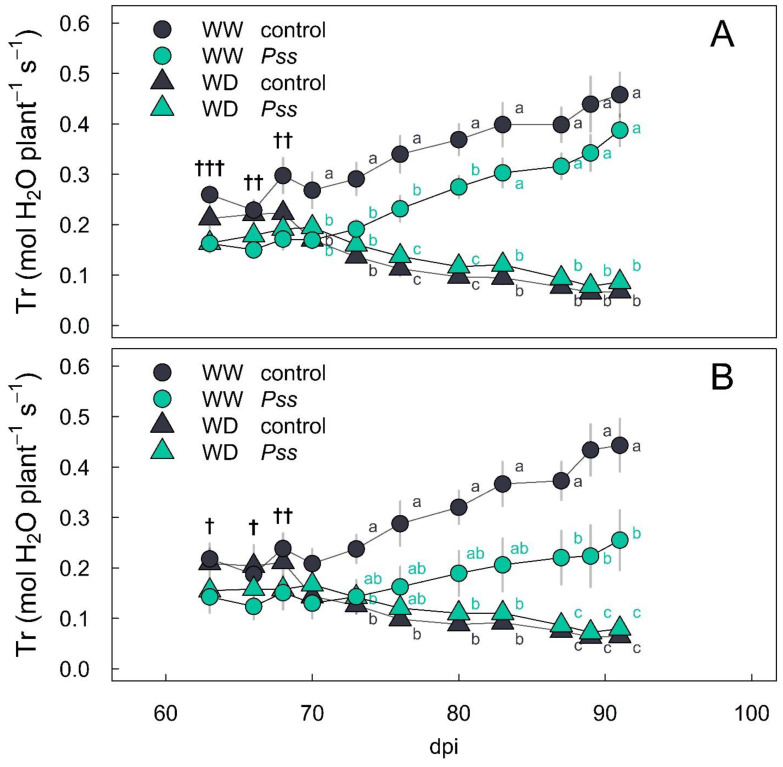
Whole-plant transpiration rate (Tr) in sweet cherry plants under sequential biotic and abiotic stress. Summer pattern of Tr in ‘Bing’ (**A**) and ‘Santina’ (**B**) plants inoculated with control (black symbols) and *Pss* (green symbols) during spring. Sweet cherry plants were subsequently exposed to well-watered (WW, circles) and water-deficit (WD, triangles) irrigation regimes. Symbols and vertical bars represent the mean ± SE (n = 6). Crosses indicate significant differences for inoculation treatment († *p* < 0.05, †† *p* < 0.01, ††† *p* < 0.001). In the case of a significant interaction (*p* < 0.05) between inoculation and irrigation, a marginal means post hoc Tukey test was performed. Different letters denote significant differences between inoculation and irrigation treatments.

**Figure 4 plants-13-01737-f004:**
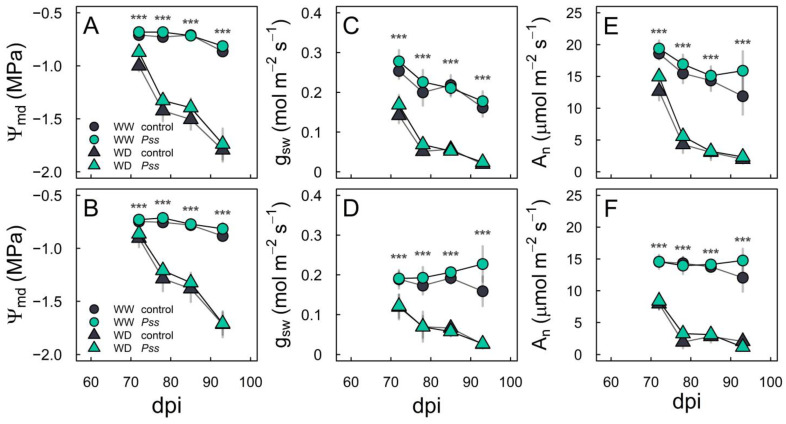
Water relations in sweet cherry plants under sequential biotic and abiotic stress. Midday leaf water potential (Ψ_md_; left panel), stomatal conductance (g_sw_; middle panel), and net assimilation (A_n_, right panel) in ‘Bing’ (**A**,**C**,**E**) and ‘Santina’ (**B**,**D**,**F**) subjected to spring inoculations with control (dark symbols) or *Pss* (green symbols) and subsequently exposed to well-watered (WW, circles) and water-deficit (WD, triangles) irrigation regimes during summer. Symbols and vertical bars represent the means ± SE (n = 6). The asterisks indicate significant differences for the irrigation regimes (*** *p* < 0.001).

**Figure 5 plants-13-01737-f005:**
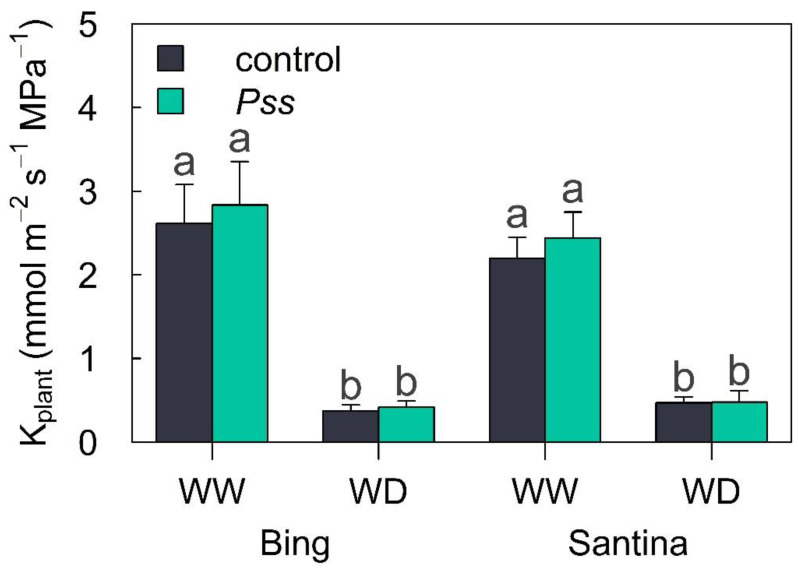
Whole-plant hydraulic conductance in sweet cherry plants under sequential biotic and abiotic stress. Bar plot showing the whole-plant hydraulic conductance (K_plant_) at 93 dpi in sweet cherry plants of Bing and Santina inoculated with the control (black) and *Pss* (green) during spring and subjected to well-watered (WW) and water deficit (WD) irrigation regimes during summer. In the case of a significant interaction (*p* < 0.05) between inoculation, cultivars, and/or irrigation, a marginal means post hoc Tukey test was performed. Different letters denote significant differences between inoculation and irrigation treatments. Different letters denote significant differences between inoculation and irrigation treatments at *p* < 0.05.

**Figure 6 plants-13-01737-f006:**
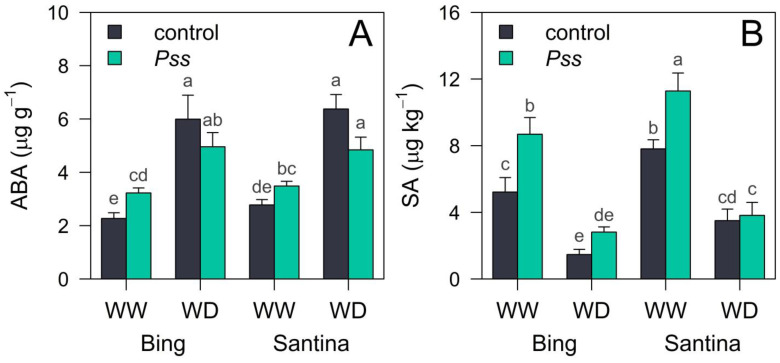
ABA and SA levels in leaves of sweet cherry plants under sequential biotic and abiotic stresses. Bar plots of (**A**) abscisic acid (ABA) and (**B**) salicylic acid (SA) levels in mature leaves at 93 dpi. ‘Bing’ and ‘Santina’ plants subjected to spring inoculations with the control (black) and *Pss* (green) and subsequently exposed to well-watered (WW) and water-deficit (WD) irrigation regimes during summer. The bars indicate means and SE (n = 6). In the case of a significant interaction (*p* < 0.05) between inoculation, cultivars, and/or irrigation, a marginal means post hoc Tukey test was performed. Different letters denote significant differences between treatments.

**Table 1 plants-13-01737-t001:** Presence of macroscopic responses to *Pss* inoculation observed in plants (n = 21) of ‘Bing’ and ‘Santina’ at 14 dpi.

	‘Bing’		‘Santina’	
Type of Response	Control	*Pss*	Control	*Pss*
no responses	21 (100%)	6 (29%)	21 (100%)	2 (10%)
local necrosis < 5 cm + gum	0	13 (62%)	0	5 (24%)
extended necrosis > 5 cm + apex death	0	2 (10%)	0	14 (67%)
total plants (each group)	21	21	21	21

**Table 2 plants-13-01737-t002:** Foliar characteristics and final biomass in sweet cherry plants after spring inoculation with *Pss* and subsequent summer exposure to different irrigation levels. Total leaf area, number of leaves, fallen leaves, root biomass and plant biomass. Mean values ± SE (n = 6).

			Leaf Area	Leaves	Leaf Abscission	Root Biomass	Plant Biomass
Irri	Ino	Var	(m^2^)	(n)	(%)	(g dw)	(g dw)
WW	control	Bing	1.01 ± 0.24 a	162 ± 24 a	10.9 ± 2.2 b	138 ± 35 a	450 ± 66 a
WW	control	Santina	1.12 ± 0.14 a	135 ± 10 b	13.1 ± 2.7 b	128 ± 36 ab	437 ± 72 a
WW	*Pss*	Bing	0.87 ± 0.2 ab	117 ± 16 bc	11.7 ± 2.5 b	106 ± 23 bc	373 ± 54 ab
WW	*Pss*	Santina	0.63 ± 0.3 b	65 ± 33 d	14 ± 3.2 b	69 ± 23 c	245 ± 113 bc
WD	control	Bing	0.61 ± 0.16 b	122 ± 13 b	9.3 ± 2.2 b	76 ± 18 c	261 ± 53 c
WD	control	Santina	0.54 ± 0.22 b	78 ± 28 cd	33.4 ± 4.3 a	83 ± 37 c	238 ± 88 c
WD	*Pss*	Bing	0.63 ± 0.17 b	128 ± 19 b	10 ± 2.3 b	65 ± 13 c	248 ± 39 c
WD	*Pss*	Santina	0.62 ± 0.06 b	71 ± 37 cd	35.2 ± 4.5 a	79 ± 21 c	274 ± 57 bc
Effects					*p*-value		
*Irri*			<0.0001	<0.0001	0.0049	<0.0001	<0.0001
*Ino*			0.018	0.0097	0.3881	0.0018	0.0064
*Var*			0.3251	0.0117	<0.0001	0.287	0.0857
*Irri × Ino*			0.0013	0.0109	n.s.	0.0077	0.0031
*Ino × Var*			0.7927	0.003	n.s.	0.0499	0.0891
*Irri × Var*			0.1536	n.s.	0.0023	n.s.	0.4543
*Irri ×* *Ino ×* *Var*			0.0659	n.s.	n.s.	n.s.	0.0441

Note. In the case of a significant interaction (*p* < 0.05) between inoculation, cultivars, and/or irrigation, a marginal means post hoc Tukey test was performed. Different letters denote significant differences between treatments for each variable at *p* < 0.05. Irri: irrigation; Ino: inoculation; Var: cutivar; n.s. denotes removal of an interaction to obtain the more parsimonious model.

## Data Availability

Data are available from the authors upon request.

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
