# Peer review of "Sweet Cherry Plants Prioritize Their Response to Cope with Summer Drought, Overshadowing the Defense Response to Pseudomonas syringae pv. syringae"

_plants, 2024, doi:10.3390/plants13131737_

Round 1

Reviewer 1 Report

Comments and Suggestions for Authors

1. The authors compared the responses of two cherry cultivars, 'Bing' and 'Santina', to Pss inoculation, including necrosis progression, gum secretion, and apex death. However, the photograph angle in Figure 1A is inappropriate, making it impossible to observe necrosis progression and apex death.

2. Pss, along with a group of Pseudomonas spp. and pathovars (P. spp. pvs.), is responsible for causing bacterial canker disease in Prunus spp. How did the authors determine that the inoculation with Pss induced bacterial canker disease in 'Bing' and 'Santina'?

3. At 14 dpi, partial death of the cherry samples was observed. However, Pss was not detected in stem segments 5 cm away from the inoculation point. Was this death caused by Pss? What could be the reason?

4. After 14 dpi, 'Bing' and 'Santina' cherry samples may have exhibited varying degrees of death. Were the remaining samples immune escape? How did the authors determine that Pss remained toxic to the remaining samples?

Author Response

Reviewer 1

  1. The authors compared the responses of two cherry cultivars, 'Bing' and 'Santina', to Pss inoculation, including necrosis progression, gum secretion, and apex death. However, the photograph angle in Figure 1A is inappropriate, making it impossible to observe necrosis progression and apex death.

Thanks for the observation.

We included images of whole plants in the supplementary material data 4. In these images you can find a picture of the control plants of Bing (A) and Santina (B), and a close-up of the healed lesion in the control plant (C), and a picture of whole plants inoculated with Pss of Bing (D) and Santina (E).

  1. Pss, along with a group of Pseudomonas spp. and pathovars (P. spp. pvs.), is responsible for causing bacterial canker disease in Prunus spp. How did the authors determine that the inoculation with Pss induced bacterial canker disease in 'Bing' and 'Santina'?

Thanks for the question.

The pathogen possesses the capability to cause mortality in both young and mature trees.

The development of cankers, which can girdle branches and entire trees, is a frequent occurrence in mature trees. These events contribute to the rapid decline of older orchards. But systemic infection leading to the death of young trees is a persistent issue in nurseries.

In our experiment, where we use young plants (one year old), we have included the following statement in the introduction (lines 53-55): ‘Pss, along with a group of Pseudomonas spp. and pathovars (P. spp. pvs.), is responsible for causing systemic infection leading to the death of young trees in nurseries and bacterial canker disease in older trees. This is one of the main constraints for Prunus spp. cultivation.’ This addition aims to provide clarity.

  1. At 14 dpi, partial death of the cherry samples was observed. However, Pss was not detected in stem segments 5 cm away from the inoculation point. Was this death caused by Pss? What could be the reason?

Thanks for the question.

The use of a moderately aggressive Pss strain and a high concentration of the bacterial inoculum in the trial, triggered a hypersensitivity response in one-year-old plants. This explain the absence of the bacteria in distal tissues, as described in discussion. For futher information, please refer to lines 321- 323.

  1. After 14 dpi, 'Bing' and 'Santina' cherry samples may have exhibited varying degrees of death. Were the remaining samples immune escape? How did the authors determine that Pss remained toxic to the remaining samples?

Thanks for the questions.

Were the remaining samples immune escape?

It is not possible to determine if the remaining tissue was immune to the bacteria, because no further inoculations were performed on the material. Performing such an experiment would undoubtedly be of great interest; however, for this trial, having performed further inoculations implied:

Suboptimal Conditions for bacterial development: The air temperature fluctuated from 10°C to 30°C, with a minimum relative humidity (%RH) of 30% and an average of 60%.

Risk of losing material: There was a risk of losing the material that had to be subsequently subjected to different levels of irrigation during the summer.

How did the authors determine that Pss remained toxic to the remaining samples?

We are unable to determine the specific toxicity of Pss in the remaining samples, but we found that the temporary presence of Pss in plants inoculated with the bacteria resulted in reduced growth compared to control plants. Such a negative effect on growth may be detrimental at the production level, especially in nurseries.

Reviewer 2 Report

Comments and Suggestions for Authors

Comments in the attached document. Please revise accordingly.

Author Response

Reviewer 2

Thanks to reviewer 2 for his thoughtful comments, which were addressed below:

L22 add cultivar name.

Accepted and changed.

L23 add in brackets which parameterts did you measure for all.

Accepted and changed.

L24 brief explanation for factors, inoculation and irrigation levels,

To maintain the length of the abstract associated with the publisher, for further details please refer to lines 79-85

L40 explain please.

Added “in agricultural systems compared to natural environments”, please refer to L43

L43 can you provide some new info, data, for specific species, especially sweet cherry?,

Added “the productivity and viability of crops such as wheat, rice, corn, and soybean and also fruit trees including as Prunus species. please refer to L46-47

L44 here you mean sweet cherry, right? Its Prunus avium... not sour cherry. Please correct.

Accepted and changed. Please refer to L48

L89 name the cultivars! Use the term cultivar, instead of variety.

Accepted and changed. Please refer to L97

L104 does any data exist on this topic in the literature? a short paragraph should be provided here related to this.

Physiological variables were changed to “water potential, stomatal conductance, photosynthesis, plant hydraulic conductance,” in L112-113, and further detail were provided in L79-85

L106 cultivar.

accepted and changed. Please refer to L115

L368 The space.

Accepted and corrected. Please refer to L391

L379 how old were the trees in your experiment?

One year old, included in L409

L391 No need for these references

References were removed.

L425 how old were threes? did you provided enough substrate for trees?

One year old, included in L409. Yes, the amount of substrate was appropriate for each plant. When the roots were controlled, they were completely submerged in the substrate used.

L427 did you measure and control the soil moisture?

Although a soil moisture measurement was not conducted, we performed a gravimetric measurement of the weight of each pot within 1 hour after sunrise. This method allowed us to control irrigation levels in advance. For further details on this topic, please consult the following references.

Opazo, I., Toro, G., Salvatierra, A., Pastenes, C., & Pimentel, P. (2020). Rootstocks modulate the physiology and growth responses to water deficit and long-term recovery in grafted stone fruit trees. Agricultural Water Management, 228, 105897. https://doi.org/10.1016/j.agwat.2019.105897

Sinclair, T. R., & Ludlow, M. M. (1986). Influence of Soil Water Supply on the Plant Water Balance of Four Tropical Grain Legumes. Functional Plant Biology, 13(3), 329–341. https://doi.org/10.1071/PP9860329

Toro, G., Pastenes, C., Salvatierra, A., & Pimientel, P. (2023). Trade-off between hydraulic sensitivity, root hydraulic conductivity and water use efficiency in grafted Prunus under water deficit. Agricultural Water Management, 282, 108284. https://doi.org/10.1016/j.agwat.2023.108284

L462 explain this term.

Accepted and changed. Please refer to the L486

L466 how did you determine that shoot? can you describe the position of the shoot, sun exposure etc...,

Accepted and added, “For this purpose, the shoot near the apex, which was completely exposed to sunlight, was selected.” Please refer to L500-501

L473 Leaf chlorophyll fluorescence,

Of course, it would have been very interesting to evaluate measurements related to chlorophyll fluorescence, but unfortunately, we did not have available to a fluorescence chamber during the experiment.

L476 other info? sun exposed leaves, shaded leaves, damaged/undamaged... etc...?? Position on the shoot??

Accepted and changed to “on fully expanded, undamaged leaves located between nodes 7 and 15 from apex to base and fully exposed to the sun.” Please refer to L510-512

L517-518 chromatogram and method,

We provide the chromatogram to the reviewer in the following picture.

Chromatogram for salicylic acid and ABA (page 1 attached document)

L577. Please provide additional paragraph what do you suggest to the sweet cherry producers for the future production of sweet cherries? suggest tips for growers!

We included the following.

“The research carried out during this study will offer critical information for both growers and nurseries on the selection of sweet cherry cultivars, the suitability of disease-free material and the adverse effects on plants even when transiently affected by the Pss. This is particularly relevant during the growth phase in young sweet cherry plants.”

Please refer to L615-619

L577. provide a short paragraph about limitations of the study and suggest future work in this field.

We include a limitation of the study.

“Limitations of the study and future work”.

In this study, one-year-old plants subjected to inoculations with a medium pathogenicity strain of Pss were used. Furthermore, the removal of tissue near the inoculation site (also that which was necrotic) introduces uncertainty as to the extent of damage caused by longer exposure to Pss with a strain of higher pathogenicity and/or the use of older plants. In future experiments, it would be useful to analyze whether cherry cultivars exhibit altered responses to Pss when previously exposed to drought conditions.”

Please refer to L398-405

Reviewer 3 Report

Comments and Suggestions for Authors

Dear authors. Thank you for the valuable research and excellent manuscript. I have only some remarks:

93 references are far too much for the manuscript, if it is not a review paper. Please, review a list of references and delete which are not necessary or duplicating each other. In many cases, one statement is supported by 3 -5 references, and some of them are not related to fruit crops and could be deleted.

Figure 2. Explain measurement unit ‘°Cd per leaf’.

Author Response

93 references are far too much for the manuscript, if it is not a review paper. Please, review a list of references and delete which are not necessary or duplicating each other. In many cases, one statement is supported by 3 -5 references, and some of them are not related to fruit crops and could be deleted.

Thank you for taking the time to review this manuscript. 

We reduced the number of references

Figure 2. Explain the measurement unit ‘°Cd per leaf’.

Thanks for the observation

The unit is °C day per leaf and it is a typical unit for growing degree days, it was included in the legend and changed in figure axis for clarity.

Please refer to L154 and L161

Round 2

Reviewer 2 Report

Comments and Suggestions for Authors

All comments have been successfully addressed.